# Daunorubicin and Its Active Metabolite Pharmacokinetic Profiles in Acute Myeloid Leukaemia Patients: A Pharmacokinetic Ancillary Study of the BIG-1 Trial

**DOI:** 10.3390/pharmaceutics14040792

**Published:** 2022-04-05

**Authors:** Guillaume Drevin, Marie Briet, Caroline Bazzoli, Emmanuel Gyan, Aline Schmidt, Hervé Dombret, Corentin Orvain, Aurelien Giltat, Christian Recher, Norbert Ifrah, Philippe Guardiola, Mathilde Hunault-Berger, Chadi Abbara

**Affiliations:** 1Service de Pharmacologie-Toxicologie et Pharmacovigilance, Centre Hospitalo-Universitaire d’Angers, F-49100 Angers, France; guillaume.drevin@chu-angers.fr (G.D.); marie.briet@chu-angers.fr (M.B.); 2UFR Santé, Université Angers, F-49100 Angers, France; alschmidt@chu-angers.fr (A.S.); noifrah@chu-angers.fr (N.I.); phguardiola@chu-angers.fr (P.G.); mahunault@chu-angers.fr (M.H.-B.); 3MITOVASC, Equipe CarMe, SFR ICAT, INSERM, CNRS, F-49000 Angers, France; 4Grenoble INP, TIMC-IMAG, Université Grenoble Alpes, CNRS, F-38000 Grenoble, France; caroline.bazzoli@univ-grenoble-alpes.fr; 5Service d’Hématologie et Thérapie Cellulaire, Equipe LNOx, ERL CNRS 7001, Centre Hospitalier Universitaire, Université de Tours, F-37000 Tours, France; e.gyan@chu-tours.fr; 6Fédération Hospitalo-Universitaire GOAL, F-49033 Angers, France; corentin.orvain@chu-angers.fr (C.O.); aurelien.giltat@chu-angers.fr (A.G.); 7Service des Maladies du Sang, Centre Hospitalo-Universitaire d’Angers, F-49100 Angers, France; 8Inserm, CRCINA, SFR ICAT, Université Angers, Université de Nantes, F-49000 Angers, France; 9Blood Disease Department, University Hospital Saint Louis AP-HP, F-75010 Paris, France; herve.dombret@aphp.fr; 10Insitut Universitaire du Cancer de Toulouse Oncolpole, Unversité Toulouse III Paul Sabatier, F-31000 Toulouse, France; recher.christian@iuct-oncopole.fr

**Keywords:** daunorubicin, pharmacokinetics, acute myeloid leukaemia, modelling

## Abstract

Daunorubicin pharmacokinetics (PK) are characterised by an important inter-individual variability, which raises questions about the optimal dose regimen in patients with acute myeloid leukaemia. The aim of the study is to assess the joint daunorubicin/daunorubicinol PK profile and to define an optimal population PK study design. Fourteen patients were enrolled in the PK ancillary study of the BIG-1 trial and 6–8 samples were taken up to 24 h after administration of the first dose of daunorubicin (90 mg/m^2^/day). Daunorubicin and daunorubicinol quantifications were assessed using a validated liquid chromatography technique coupled with a fluorescence detector method. Data were analysed using a non-compartmental approach and non-linear mixed effects modelling. Optimal sampling strategy was proposed using the R function PFIM. The median daunorubicin and daunorubicinol AUC0-tlast were 577 ng/mL·hr (Range: 375–1167) and 2200 ng/mL·hr (range: 933–4683), respectively. The median metabolic ratio was 0.32 (range: 0.1–0.44). Daunorubicin PK was best described by a three-compartment parent, two-compartment metabolite model, with a double first-order transformation of daunorubicin to metabolite. Body surface area and plasma creatinine had a significant impact on the daunorubicin and daunorubicinol PK. A practical optimal population design has been derived from this model with five sampling times per subject (0.5, 0.75, 2, 9, 24 h) and this can be used for a future population PK study.

## 1. Introduction

With the exception of acute promyelocytic leukaemia, current care practices for acute myeloid leukaemia (AML) patients in good condition consist of an induction phase combining cytarabine and anthracycline—daunorubicin or idarubicin—followed by a post-induction phase relying on cytarabine administration. The gold-standard ‘3 + 7’ induction regimen associates cytarabine from day 1 to day 7 and daunorubicin or idarubicin from day 1 to day 3 [1].

Daunorubicin is a cytotoxic agent acting through a topoisomerase-mediated interaction with deoxyribonucleic acid (DNA), thereby inhibiting DNA replication and repair as well as ribonucleic acid (RNA) and protein synthesis. Daunorubicin is extensively metabolised into a hydroxyl derivative, daunorubicinol—which is considered to be less cytotoxic—through a complex process involving both carbonyl reductases 1 (CBR1) and 3 (CBR3) [2]. Daunorubicin pharmacokinetics (PK) are characterised by a large inter-individual variability that may impact treatment efficacy and the outcome of AML patients. Several randomised clinical trials have demonstrated that increasing the doses of anthracyclines during the induction phase significantly improved the outcomes of AML patients [3,4]. In addition, inter-individual variability in daunorubicin metabolism can result in potential complications regarding toxicity, as well as efficacy [5,6,7]. These observations raise questions about the optimal dose regimen for daunorubicin administration and daunorubicinol contribution with respect to the clinical outcome. A pharmacokinetic/pharmacodynamic (PK/PD) population study is a strategy of choice to help define the optimal dose regimen for this cytotoxic agent. This approach was preferred to physiologically based PK, as it provides a quantitative framework to account for inter-individual variability in drug exposure, and the influence of covariates [8]. However, conducting such a study first requires the establishment of a joint daunorubicin/daunorubicinol PK model following daunorubicin administration, according to current schemes.

Based on this assumption, we performed a prospective PK evaluation of daunorubicin/daunorubicinol following its administration as a 30-min IV infusion at 90 mg/m^2^ in AML patients included in the phase III/II prospective BIG-1 clinical trial and developed a joint daunorubicin–daunorubicinol structural pharmacokinetic model.

## 2. Materials and Methods

### 2.1. Patients

This study is an ancillary PK study of the BIG-1 clinical trial (NCT02416388). BIG-1 is a multicentre phase III/II prospective clinical trial aiming to improve the overall survival rate of AML patients aged between 18 and 60 years (excluding AML-M3). It combines multiple randomisation processes in order to address questions aimed at improving the outcome of those patients, including a comparison of high-dose idarubicin (9 mg/m^2^/day, from day 1 to day 5) to high-dose daunorubicin (90 mg/m^2^/day from day 1 to day 3) during the induction phase, and a comparison of high-dose to intermediate-dose cytarabine during the post-induction phase.

From May 2018 to December 2019, 14 patients who were receiving daunorubicin during their induction phase were included in the PK ancillary study. The therapeutic scheme was daunorubicin 90 mg/m^2^/day from day 1 to day 3, 30 min intravenous infusion, with cytarabine 200 mg/m^2^/day from day 1 to day 7 as a continuous 24-h intravenous infusion.

Signed written informed consent was obtained for each patient included in this study, which was approved by the “comité de protection des personnes/OUEST II” (2014-000699-24, 28 March 2018).

For each of these patients, the following clinical and biological covariates were collected at diagnosis: age, gender, body surface area (BSA), co-medications, cytogenetic risk group, cytological FAB-classification, prognostic molecular markers (FLT3, NPM1, CEBPA), white blood cell (WBC) count, platelet count, blood blast percentage, bone marrow blast percentage, peripheral blood lymphocyte count, haemoglobin level, C-reactive protein (CRP), plasma creatinine and prothrombin time.

### 2.2. Pharmacokinetic Sampling

Blood samples were collected in lithium heparin tubes prior to daunorubicin infusion, and at 5 min, 0.25, 0.5, 0.75, 1, 2, 4, 6, 12 and 24 h after the end of daunorubicin infusion on the first day of the induction phase. The plasma was separated by centrifuging the sample at 3500× *g* for 5 min and stored at −20 °C until analysis. The maximum delay between blood drawing and centrifugation/storage was 30 min.

### 2.3. Plasma Quantitation of Daunorubicin and Daunorubicinol

Daunorubicin and daunorubicinol plasma quantifications were performed by means of high-performance liquid chromatography (HPLC) coupled with a fluorescence detector (FLD). This is a bioanalytical method developed in-house. Briefly, plasma samples were treated with ammonium acetate buffer (pH 9) and chloroform/isopropyl alcohol using doxorubicin as internal standard (IS) (2 mg/L). After mixing the preparation and centrifugation (3500× *g* for 5 min), the non-aqueous phase was evaporated at 37 °C under nitrogen gas. The residue was dissolved in 30 µL of acetonitrile then completed to a total volume of 100 µL with water, mixed and centrifuged (3500× *g* for 5 min). The supernatant was injected into the HPLC-FLD system. Chromatographic analysis was carried out using an Agilent 1260 HPLC system (Agilent Technologies, Les Ulis, France). Separation was achieved using an Uptisphere C18 ODB column (5 µm, 100 × 2.1 mm). The mobile phase was composed of a mixture of ammonium acetate buffer (pH 4) and acetonitrile. The gradient solvent (flow-rate 0.4 mL/min) was as follows: 70/30 (*v*/*v*) at time 0 for 6 min, to 20/80 at 6.5 min for a further 5 min before return to initial conditions. The injection volume was 10 µL and the total run time was 11.5 min. The retention times of daunorubicin, daunorubicinol and the IS were 3.8, 2.4 and 1.8 min, respectively. The validation of the analytical method was based on the Food and Drug Administration (FDA) guidelines for bioanalytical method validation [9]. The calibration curve was established using concentrations ranging from 10 to 1000 µg/L with a linear regression and 1/concentration weighing model with a determination coefficient >0.99. The lower limit of quantitation (LLOQ) was 10 ng/mL. Within-run and between runs validation results are summarised in Appendix A. The results of the stability study are presented in Appendix A.

### 2.4. Pharmacokinetic Analyses

#### 2.4.1. Non-Compartmental Analysis

A non-compartmental analysis (NCA) of daunorubicin and daunorubicinol concentration data was performed using Pkanalix (Lixoft, Orsay, France). Concentrations below the LOQ were replaced by the LLOQ/2 so as not to lose information [10]. AUC from time 0 to the last sampling time (AUC_0-tlast_) and peak plasma concentrations (C_max_) were the parameters of interest. AUC was calculated using a log-linear trapezoidal method. The metabolic ratio (Daunorubicin AUC_0-tlast_/metabolite AUC_0-tlast_) was also calculated. Univariate linear regression analyses were performed to evaluate the relationship between NCA PK parameters, described previously, and each clinical and biological covariate collected at diagnosis. The categorical and continuous covariates tested were: age, sex, body surface area (BSA), cytogenetic risk group, FAB classification, WBC count, platelets, blood blast percentage, bone marrow blast percentage, blood lymphocyte count, haemoglobin, C-reactive protein (CRP), plasma creatinine and prothrombin time. A multivariate stepwise linear regression model was then performed including the covariates of the patients previously described with *p*-values below 0.30 in univariate analyses.

#### 2.4.2. Joint Parent-Metabolite Pharmacokinetic Model Development and Evaluation

Daunorubicin and daunorubicinol concentration data were analysed using a non-linear mixed effect model. The stochastic approximation expectation maximisation (SAEM) algorithm implemented in the MonolixSuite 2019R2 software (Lixoft, Orsay, France) was used to estimate the parameters. Plasma concentrations below the LLOQ were censored in accordance with the method described by Beal [10]. Structural models were built using user-defined ODE functions written in the Mlxtran language. The structural models tested were two and three-compartment parent-metabolite models with first-order transformation of daunorubicin to its metabolite, without back transformation and first-order elimination for both molecules. Pharmacokinetic parameters were log-normally distributed. The inter-individual variability was described by an exponential model. Three models of residual error were tested: additive, proportional and combined.

The impact of the clinical and biological covariates collected at diagnosis was evaluated in order to determine their influence on inter-individual variability. Only covariates with a significant effect on PK NCA parameters in multivariate analyses were considered. The influence of continuous covariates was modelled as follows:ln(θ_i_) = ln(θ_pop_) + β_cov_ · cov_i_
where θ_pop_ is the typical value of the parameter in the population, θ_i_ the value of the individual parameter influenced by the covariate cov_i_, and β_cov_ is the coefficient associated to the covariate.

The COnditional Sampling for Stepwise Approach based on Correlation tests (COSSAC) automatic procedure implemented in Monolix^®^ was used for selecting covariates. Covariate integration was based on −2 LL (forward *p* = 0.01; backward *p* = 0.001) and correlation > 0.3. The final choice of covariates was based on the results of COSSAC and their clinical relevance.

The model development was guided by the minimum value of the corrected Bayesian Information Criteria (BICc), which is penalised by the log of the number of subjects and the log of the total number of observations. The standard errors for the estimated population parameters were calculated via the estimation of the Fisher information matrix. An internal model validation was performed through goodness-of-fit graphs. Simulation-based diagnostics were conducted using a prediction-corrected Visual Predictive Check (pcVPC).

#### 2.4.3. Design Optimisation for a Future Population PK Study of Daunorubicin/Daunorubicinol

Based on the basic joint model developed and its estimated parameters, we optimised a population design with only five sampling times per patient among a set of 17 admissible sampling times (0.5, 0.75, 1, 1.5, 2, 2.5, 3, 4, 5, 6, 7, 8, 9, 10, 12, 18 and 24 h) for a total number of samples equal to 200 for both molecules, i.e., the total number of subjects equal to 20. For this purpose, the R function PFIM 4.0.2 was adapted [11] for a population PK study design with multiple responses. It assessed the expected standard errors for the parameter estimates for the design evaluated or optimised. Optimisation was performed using the Federov–Wynn algorithm, which provides an optimal design described by means of a number of elementary designs with the corresponding sampling times and proportion of subjects in each elementary design.

## 3. Results

### 3.1. Population Description

Clinical and biological characteristics of the patients included, at diagnosis, are presented in Table 1.

Median age was 48 years (range, 32–60 years). Five patients were females and nine were male. The median administered dose of daunorubicin was 177 mg/day (range, 131–232 mg/day). The median administered dose of cytarabine was 390 mg/day (range, 290–515 mg/day). The biological characteristics of each AML (cytology, cytogenetics and molecular markers) are detailed in Appendix A. The co-medications are reported in Appendix A.

### 3.2. Non-Compartmental Analysis

The complete dataset of 14 patients included 132 concentration values for daunorubicin and its metabolite. Sixteen concentrations were below the LLOQ for daunorubicin and none for daunorubicinol. Individual concentration profiles are presented as spaghetti plots in Figure 1.

NCA results are presented in Appendix A. Multivariate analyses did not identify any significant relationship between daunorubicin AUC_0-tlast_ and the clinical and biological covariates of the patients considered. The combination of plasma creatinine, BSA, WBC count, peripheral blood lymphocytes count, blood blast percentage, bone marrow blast percentage and prothrombin time explained a large amount of the variability observed in the daunorubicinol AUC_0-tlast_ (R^2^ = 0.982). Finally, the metabolic ratio was significantly linked to a combination of plasma creatinine, WBC count, cytogenetic risk and peripheral blood blast percentage (R^2^ = 0.905).

### 3.3. Joint Population Parent-Metabolite PK Model

The structural model that best described the PK of daunorubicin and its metabolite was a three-compartment parent (central compartment: V1; 1st peripheral compartment: V2; 2nd peripheral compartment: V3; intercompartmental 1 and 2 clearance: Q1; intercompartmental 1 and 3 clearance: Q2) and a two-compartment metabolite model (central metabolite compartment: V4; peripheral metabolite compartment: V5; intercompartmental clearance: Q3) (Figure 2).

Two transformation rates were used to describe the PK of daunorubicinol, one from the central parent compartment to the central metabolite compartment (kp1m) and another from the second peripheral parent compartment to the peripheral metabolite compartment (kp3m) and a first-order elimination of daunorubicinol (metabolite clearance Cl_m_). Daunorubicin elimination from the central compartment could not be identified, as the value of the estimated elimination constant was extremely small (<0.001 L/h) and poorly estimated (RSE > 250%). Volumes of distribution of central compartments of both parent and metabolite were considered as equal (V1 = V4) (Appendix B). The error model that best fit the data was the proportional model for both substances. Inter-individual variability was removed on two parameters, V2 and kp3m, and a substantial random effect was detected for Cl_m_ and Q3. A significant correlation was observed between the inter-individual variability of Q1 and Q2. The effects of the following covariates were then evaluated: plasma creatinine, BSA, peripheral blood blast percentage, WBC count, peripheral blood lymphocytes count and prothrombin time. BSA had a significant effect on Cl_m_ and V5 and plasma creatinine had a significant impact on Cl_m_. In the final model, the shrinkage accounted for less than 20% of all fixed effects for which an inter-individual variability was estimated (Table 2). The final equations for the individual estimation of parameters impacted by covariates are:ln(V5_i_) = ln(79.2) + 0.98 × BSA_i_
ln(Cl_mi_) = ln(35.8) − 0.027 × creatinine_i_ + 1.04 × BSA_i_
where V5_i_ and Cl_mi_ are the estimations of the parameters V5 and Cl_m_ for the individual i, BSA_i_ and creatinine_i_ are the values of BSA and creatinine for the individual i.

PK parameter estimates for both basic and full models are presented in Table 2.

The evaluation of goodness-of-fit plots demonstrated that residuals exhibited no apparent trend (Figure 3).

According to the VPC (Figure 4), the average observed values were well predicted, and inter-individual variability was somewhat overestimated for daunorubicin.

The model developed fitted the individual data well. Individual predicted profiles using the model described above are shown in the Appendix A.

### 3.4. Design Optimisation for a Future Population PK Study

The optimal designs obtained by PFIM are detailed in Appendix A. For the same total number of samples as the initial design and the possibility of subgroups with different elementary designs, the best optimal design D_opt_ was composed of five groups, including between one and seven subjects with the same sampling time points (*n* = 5) for both molecules in each group. This design returned good relative standard errors (RSE) (<30%) for fixed parameters. However, as this scheme can be difficult to apply in clinical practice, a one-group design with the same sampling time points for both molecules and all patients was evaluated. The best one group design (D_opt one_) is detailed in Appendix A and shown in Appendix A. The estimation of fixed-effect parameters was substantially degraded compared to the optimal design, but their respective predicted RSEs were still below 30%.

## 4. Discussion

In this study, we described a joint population PK model for daunorubicin and its metabolite daunorubicinol after its standard administration as a 30-min infusion at a dose of 90 mg/m^2^ in adult AML patients. We provided a sampling design for a population PK study that will be the first to integrate a daunorubicinol contribution.

Previous studies have described PK modelling of daunorubicin and its metabolite in adults and children but with different dosage regimens [3,6,7,12,13,14,15,16,17]. The present study provided a joint PK model of daunorubicin, and its metabolite in the current dosage scheme of daunorubicin is considered the most effective one to achieve a complete remission and to improve the outcome of adult AML patients [18]. The model developed was based on a non-linear mixed effect analysis. This approach was preferred to the physiologically based PK modelling, as the final objective of this study is the development of a structural model and deriving an optimal population design to be used in a future population PK/PD study.

The PK model that best described the experimental data integrates three-parent compartments and two-metabolite compartments. This model has two differences when compared to those previously published: (i) A tissue distribution of daunorubicin using two peripheral compartments instead of one; (ii) a second route of daunorubicin transformation to its metabolite from the second peripheral compartment. Adding the third compartment to describe daunorubicin PK was required to improve the fit of the terminal phase and to describe the second concentration peak observed in the metabolite profiles. In a three-compartment mammillary model, the shallow compartment represents drugs in rapidly equilibrated tissues (e.g., hepatic cells), and the deep compartment represents drugs in slowly equilibrated tissues (e.g., bone marrow) [19]. The part metabolised in hepatic cells is equilibrated with the central compartment, while the non-bio-transformed part is eliminated via bile. No clearance constant was identified in our model, probably due to the very small part eliminated from the central compartment by mechanisms other than metabolism (i.e., renal clearance).

The parent model (three compartments), but not the metabolite model, was in line with the model reported by Callies et al. [12]. When the Callies model [12], which involved one metabolism rate, was applied to our data, the data were not in line. In our model, adding a transformation rate from the second peripheral compartment of daunorubicin to the peripheral compartment of daunorubicinol greatly improved the fit of the second plasma daunorubicinol peak, as is demonstrated by the individual predicted profiles graph. The mechanistic interpretation can be related to the extant metabolism of daunorubicin in several tissues. This hypothesis was strengthened by the covariates effect analysis. Indeed, for both NCA and the modelling results, BSA showed a high impact on the transformation of daunorubicin into daunorubicinol. Indeed, the major enzymes involved in the metabolism of daunorubicin to daunorubicinol are carbonyl reductase 1 (CBR1) and carbonyl reductase 3 (CBR 3), which are widely expressed in a large number of organs including skin, intestine, lungs and kidneys. Therefore, the large expression of both enzymes in several tissues could explain the second plasma daunorubicinol peak as well as the second metabolism rate between the second peripheral parent compartment and the peripheral metabolite compartment integrated in our model. The clearance constant from the metabolite central compartment expresses all elimination routes of the metabolite, including renal elimination. The covariate analysis demonstrated that plasma creatinine had a significant impact on the clearance of daunorobicinol, in addition to BSA. This can be explained by the fact that only a part of daunorobicinol elimination is mediated by the kidneys. Indeed, a recent clinical study demonstrated that the daunorubicin dose fraction recovered in urine up to 144 h was 4.4% as the unchanged drug and 7.91% as daunorobicinol [7].

In the final model, it was necessary to integrate the covariates with parsimony. Hence, only covariates of clinical interest and which did not result in a large increase in RSE% values were considered in the final full model. Estimation of the RSE% increase was observed for both parameters for which a covariate effect was added (Cl_m_, V5), as is shown in Table 2. The overestimation observed in the terminal phase of daunorubicin VPC is mainly due to censored data (concentration < LLOQ).

The second aim of this study was to assess an optimal experimental design for a population PK study. Design optimisation can be of great advantage in clinical practice to avoid unfeasible sampling time points. The basic model was chosen for this determination, as predicting the AUCs of daunorubicin and daunorubicinol compared with NCA results on observed data was excellent using a covariate-free structural model. The optimised population design included five identical sampling time points per patient for both molecules. Based on these findings, this strategy will be used to evaluate the exposure response after the administration of daunorubicin in a large population of adult AML patients treated in the context of the BIG-1 clinical trial. This evaluation will be the first to consider the daunorubicinol contribution with respect to the outcome of adult AML patients receiving daunorubicin.

## 5. Conclusions

In conclusion, we have presented a suitable joint PK model of daunorubicin and daunorubicinol and the determination of an optimised experimental design for a future population PK study. This sampling strategy will be the basis for an exposure–response study that will integrate both molecules, as well as the contribution of daunorubicin and daunorubicinol to the therapeutic and toxic effects of this drug used to treat adult AML patients.

## Figures and Tables

**Figure 1 pharmaceutics-14-00792-f001:**
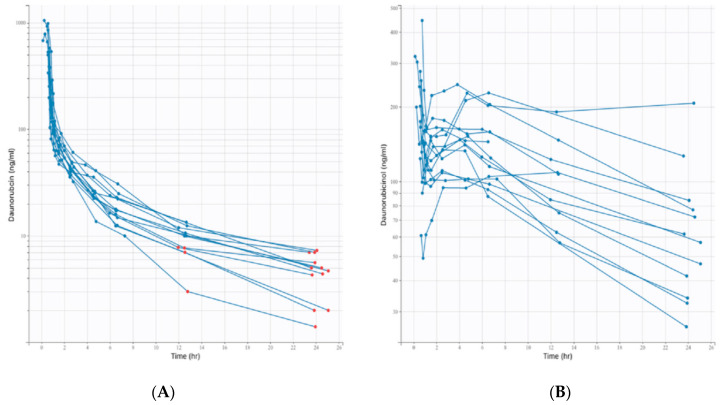
Semi-logarithmic spaghetti plots of individual PK profiles of daunorubicin (**A**) and daunorubicinol (**B**) plasma concentrations. Red dots are censored data (<LLOQ).

**Figure 2 pharmaceutics-14-00792-f002:**
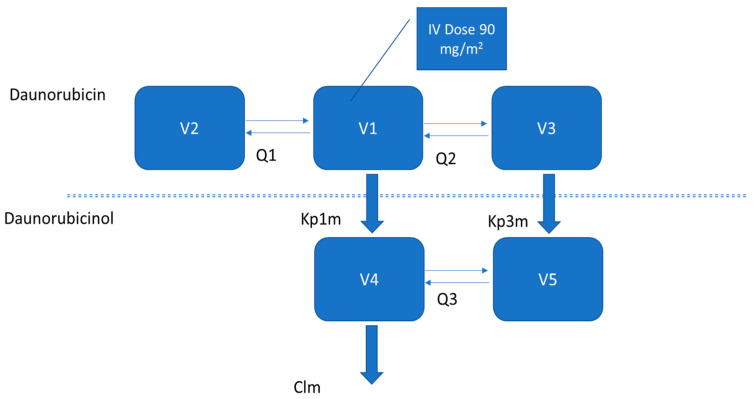
Structural PK model of daunorubicin and its metabolite daunorubicinol. V1: central parent compartment; V2: 1st peripheral parent compartment; V3: 2nd peripheral parent compartment; Q1: intercompartmental 1 and 2 clearance; Q2: intercompartmental 1 and 3 clearance; V4: central metabolite compartment; V5: peripheral metabolite compartment; Q3: intercompartmental 4 and 5 clearance; kp1m: transformation rate from V1 to V4; kp3m: transformation rate from V3 to V5.

**Figure 3 pharmaceutics-14-00792-f003:**
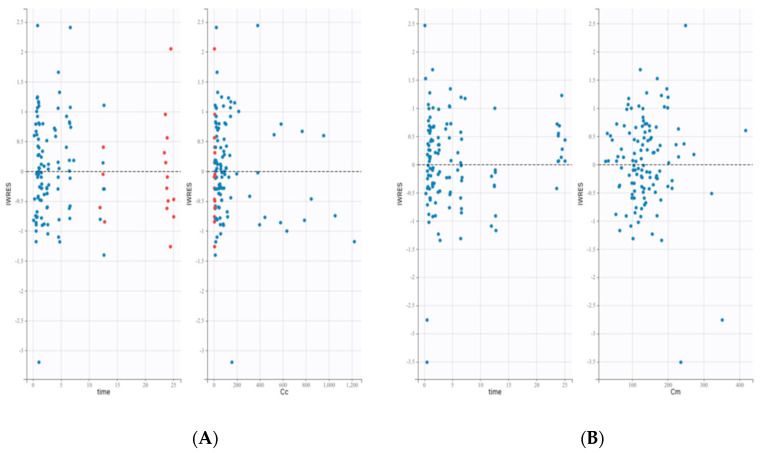
(**A**): daunorubicin residual scatterplots. (**B**): daunorubicinol residual scatterplots. These plots display the individual weighted residuals (IWRES) as scatter plots with respect to the time and the concentration. Red dots are censored data.

**Figure 4 pharmaceutics-14-00792-f004:**
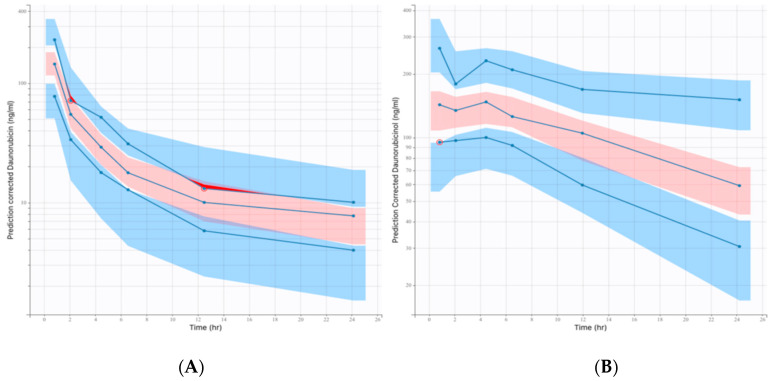
Prediction-corrected visual predictive checks (pcVPC) graphs. The red-shaded area is the 95% confidence interval (CI) of the median prediction. The blue-shaded areas are 10–90th percentiles of prediction interval. Blue curves are the empirical percentiles. (**A**) pcVPC of the final full PK model for daunorubicin concentration evolution over time; (**B**) pcVPC of the final full PK model for daunorubicinol concentration evolution over time.

**Table 1 pharmaceutics-14-00792-t001:** Clinical and biological characteristics of the 14 included patients.

**Parameters**	
Age, median (range), years	48 (32–60)
Sex, male/female, *n*(%)	9 (64%)/5 (36%)
Body surface area, m^2^	1.9 (1.3–2.5)
**Cytogenetic risk group**	
Favourable	1 (7%)
Intermediate	9 (64%)
Adverse	4 (29%)
**FAB classification**	
M1	1 (7%)
M2	7 (50%)
M5	5 (36%)
Unknown	1 (7%)
**Molecular characteristics**	
CEBPA	1 (7%)
NPM1	6 (43%)
FLT3-ITD	3 (21%)
FLT3-TKD	2 (14%)
**Biological parameters**	
WBC, median (range), Giga/L	28.6 (0.9–173.6)
Platelet, median (range), Giga/L	55.2 (27–90)
Peripheral Blasts, median (range), %	32 (0.9–91.9)
Heamoglobin, median (range), g/dL	9.7 (5.5–14.1)
CRP, median (range), mg/L	77 (2–299)
Creatinine, median (range), µmol/L	70 (56–115)
Prothrombin time, median (range), %	79 (53–93)

FAB classification: French American British classification; WBC: white blood cells; CRP: C-reactive protein.

**Table 2 pharmaceutics-14-00792-t002:** Population pharmacokinetic parameter estimates for the basic and final model.

Parameters	Population Parameter Estimates	
	Value (RSE%) Final Model	Value (RSE%) Basic Model
**Fixed effects**		
V1 (L)	22.37 (18.1)	21.1 (15.5)
V2 (L)	1393 (14.4)	1449 (14.0)
V3 (L)	330 (16.6)	323 (15.3)
Q1 (L/h)	75.1 (15.5)	69.4 (13.1)
Q2 (L/h)	135 (16.4)	125 (16.5)
Q3 (L/h)	573 (22.7)	591 (12.0)
V5 (L)	79.2 (35.7)	536 (12.0)
β BSA (m^2^) on V5	0.98 (18.3)	-
Kp1m (1/h)	3.9 (15.7)	3.73 (16.0)
Cl_m_ (L/h)	35.8 (55.6)	41.3 (16.0)
β creatinine (µmol/L) on Cl_m_	−0.027 (23.3)	-
β BSA (m^2^) on Cl_m_	1.04 (25.5)	-
Kp3m (1/h)	0.26 (6.47)	0.25 (11.0)
**Inter-individual variability**		
ω V1	36% (28.6)	40% (30.5)
ω V3	53% (29.8)	42% (26.7)
ω Q1	33% (25.2)	42% (26.0)
ω Q2	43% (25.2)	53% (24.9)
ω Q3	64% (25.5)	66% (24.5)
ω Kp1m	42% (21.9)	39% (24.1)
ω Cl_m_	30% (22.7)	58% (20.5)
ω V5	21% (25.6)	41% (19.4)
**Correlations between random effects**		
Corr. Q2 Q1	1 (0.690)	1 (0.322)
**Error model parameters**		
σ daunorubicin, b_1_	16% (8.76)	17% (9.42)
σ daunorubicinol, b_2_	11% (9.22)	11% (8.84)

Kp1m: metabolite transformation rate from V1 to V4; Cl_m_: metabolite clearance; BSA: body surface area; V1: volume of central parent and metabolite compartment; V2: volume of 1st peripheral parent compartment; V3: volume of 2nd peripheral parent compartment; Q1: intercompartmental 1 and 2 clearance; Q2: intercompartmental 1 and 3 clearance; V4: volume of central metabolite compartment; V5: volume of peripheral metabolite compartment; Q3: intercompartmental 4 and 5 clearance; kp3m: metabolite transformation rate from V3 to V5; β BSA (m^2^) on V5: coefficient associated with effect of BSA on V5; β BSA (m^2^) on Cl_m_: coefficient associated with effect of BSA on Cl_m_; β creatinine (µmol/L) on Cl_m_: coefficient associated with effect of creatinine on Cl_m_; ω: interindividual variability; σ daunorubicin, b_1_: proportional error for daunorubicin concentrations; σ daunorubicinol, b_2_: proportional error for daunorubicinol concentrations.

## Data Availability

The data presented in this study are available on request from the corresponding author. The request should be accompanied by a research protocol. The data are not publicly available due to European ethical and legal restrictions.

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
