# Peer review of "Daunorubicin and Its Active Metabolite Pharmacokinetic Profiles in Acute Myeloid Leukaemia Patients: A Pharmacokinetic Ancillary Study of the BIG-1 Trial"

_pharmaceutics, 2022, doi:10.3390/pharmaceutics14040792_

Round 1
Reviewer 1 Report
I appreciate my consideration as a reviewer regarding the manuscript “DAUNORUBICIN AND ITS ACTIVE METABOLITE PHARMACOKINETIC PROFILES IN YOUNG ACUTE MYELOID LEUKEMIA PATIENTS: A PHARMACOKINETIC ANCILLARY STUDY OF THE BIG-1 TRIAL”. This work is an interesting and well designed study of daunorubicin and its active metabolite.
I have some minor comments:
- Why did you perform NCA and PopPK analysis? I think NCA might be a good starting point, but the presentation of both methods is not needed. Could you please either delete the NCA part (or at least move it to the Supplements) or argue why you included the NCA?
- The discussion is quite long. Please shorten (especially the parts when you compare the results with previous studies are extensive).
- Could you please publish the model code? This might be a starting point for future researchers.
- RSEs of Q3 is very high, CLm at least borderline high. How do you justify these high values? Either refine your model! Or discuss these RSE-values in the discussion.
- VPC is indeed overestimating the IIV!! How are you justifying this? Please comment.
- Table 1: Abbreviations are not explained.
- Line 97: 0.083h is not intuitive, please use Minutes as a unit for the first values, then switch to hours (as it is).
- The introduction is lacking the information why the investigation of the metabolite is of interest. Is there any toxic or active function? Please add this point to the introduction.
Reviewer 2 Report
The authors present results of a pharmacokinetic study of daunorubicin and daunorubicinol. The study is part of larger trial (BIG-1 trial). The work is interesting and tries to describe the PK profile of the drug and metabolite in AML patients and furthermore develop a PK-model to analyze the results. Some comments to be said for the study is some additional info would be useful in the introduction section. For example, in lines 62-65. Why a PK/PD study is the strategy of choice and not a pharmacometric/PBPK approach? T (doi: 10.1007/s11095-018-2456-8.; doi: 10.21037/tcr.2017.09.14;). Maybe some relative references would be useful to support these statements since several works have tried to establish PK models for this drug. Moreover, are the 14 patients a sufficient number of participants to describe the PK for daunorubicin and its metabolite? Since this work is an ancillary study if they had employed additional participants would add to the results and the model?
In addition, is the HPLC method novel or based on previous one? We do not see any references thus validation parameters would be needed (in supplementary) including short-long term stability of samples prior to the analysis. Finally, a minor comment in Table 1, BSA values are missing.
